# Altered Functional Connectivity of Temporoparietal Lobe in Obstructive Sleep Apnea: A Resting-State fNIRS Study

**DOI:** 10.3390/bioengineering11040389

**Published:** 2024-04-18

**Authors:** Fang Xiao, Minghui Liu, Yalin Wang, Ligang Zhou, Jingchun Luo, Chen Chen, Wei Chen

**Affiliations:** 1School of Information Science and Technology, Fudan University, Shanghai 200437, China; 21210720039@m.fudan.edu.cn (F.X.); 21110720113@m.fudan.edu.cn (M.L.); 21110720086@m.fudan.edu.cn (L.Z.); 2School of Information Science and Engineering, Lanzhou University, Lanzhou 730000, China; 3Human Phenome Institute, Fudan University, Shanghai 200437, China; luojc@fudan.edu.cn

**Keywords:** functional near-infrared spectroscopy, resting state, obstructive sleep apnea, functional connectivity, brain network

## Abstract

Obstructive Sleep Apnea (OSA), a sleep disorder with high prevalence, is normally accompanied by affective, autonomic, and cognitive abnormalities, and is deemed to be linked to functional brain alterations. To investigate alterations in brain functional connectivity properties in patients with OSA, a comparative analysis of global and local topological properties of brain networks was conducted between patients with OSA and healthy controls (HCs), utilizing functional near-infrared spectroscopy (fNIRS) imaging. A total of 148 patients with OSA and 150 healthy individuals were involved. Firstly, quantitative alterations in blood oxygen concentration, changes in functional connectivity, and variations in graph theory-based network topological characteristics were assessed. Then, with Mann–Whitney statistics, this study compared whether there are significant differences in the above characteristics between patients with OSA and HCs. Lastly, the study further examined the correlation between the altered characteristics and the apnea hypopnea index (AHI) using linear regression. Results revealed a higher mean and standard deviation of hemoglobin concentration in the superior temporal gyrus among patients with OSA compared to HCs. Resting-state functional connectivity (RSFC) exhibited a slight increase between the superior temporal gyrus and other specific areas in patients with OSA. Notably, neither patients with OSA nor HCs demonstrated significant small-world network properties. Patients with OSA displayed an elevated clustering coefficient (*p* < 0.05) and local efficiency (*p* < 0.05). Additionally, patients with OSA exhibited a tendency towards increased nodal betweenness centrality (*p* < 0.05) and degree centrality (*p* < 0.05) in the right supramarginal gyrus, as well as a trend towards higher betweenness centrality (*p* < 0.05) in the right precentral gyrus. The results of multiple linear regressions indicate that the influence of the AHI on RSFC between the right precentral gyrus and right superior temporal gyrus (*p* < 0.05), as well as between the right precentral gyrus and right supramarginal gyrus (*p* < 0.05), are statistically significant. These findings suggest that OSA may compromise functional brain connectivity and network topological properties in affected individuals, serving as a potential neurological mechanism underlying the observed abnormalities in brain function associated with OSA.

## 1. Introduction

Obstructive Sleep Apnea (OSA) is a sleep disorder primarily characterized by repeated apnea or hypopnea during sleep. Apnea during sleep is usually due to relaxation or collapse of the throat’s tissue, resulting in an airway obstruction that prevents air from flowing into the lungs [1]. Patients with OSA often wake up during the night, even though they may not be aware of it, which can lead to poor sleep quality and cause daytime fatigue and lethargy. As a result of poor sleep quality, individuals with OSA may find it challenging to concentrate and even experience mental health issues such as depression and anxiety [2]. In addition, OSA can lead to a variety of health problems, including hypertension, heart disease, stroke, diabetes, and Alzheimer’s disease (AD) [3,4,5,6,7].

As OSA is normally accompanied by affective, autonomic, and cognitive abnormalities, numerous research studies have explored the structural changes or functional changes in the brain [8,9]. For structural brain alterations, it has been found that OSA can result in changes to white or gray matter (the two basic tissue types that make up the brain and spinal cord), which may affect the transfer of information and communication between brain regions [10,11,12]. In addition, a diffusion tensor imaging (DTI) study showed that white matter exhibited extensive alterations in individuals with OSA, affecting axonal connections among key structures within the limbic system, and frontal, temporal, and parietal cortices [13]. Studies indicate a significant correlation between the apnea hypopnea index (AHI) and white matter changes, as well as white matter hyperintensities, across different populations [14,15]. OSA leads to intermittent hypoxemia and fragmented sleep, both of which may have negative effects on gray and white matter in various regions of the brain. These adverse effects are potential contributors to memory impairments and executive function deficits, as well as dysregulation of the autonomic nervous system and respiratory control observed in individuals with OSA [10,16]. With continued treatment, these effects and patterns may be altered. Several studies have indicated that continuous positive airway pressure (CPAP) treatment can lead to adaptive changes in the neurocognitive structure of individuals with OSA [17,18]. The above investigations of structural brain alterations in patients with OSA commonly employ structural magnetic resonance imaging (sMRI). However, sMRI, applying static images, lacks the capacity to directly furnish spatiotemporal dynamic insights into brain function.

To investigate the spatiotemporal dynamics of the brain, functional brain alterations, especially abnormalities in brain functional connectivity, have been explored. Functional connectivity, where temporal correlation or synergistic activity between different functionally related brain regions at a specific period can be quantified, has been widely explored in many neuropsychiatric disorders, such as schizophrenia, depression, cognitive disorders, etc. [19,20,21]. OSA may impair cognitive function, including attention, memory, and executive function. Thus, OSA is deemed to be linked by affecting functional brain connectivity in different regions, such as the prefrontal lobes, insula, and hippocampus [22,23,24,25]. The aberrant functional connectivity (FC) observed in the sensorimotor network and default mode network (DMN) is closely associated with the neurocognitive impairments observed in individuals with OSA [22,23]. The results of abnormal FC in the hippocampus and amygdala subregions also provide a new neuroimaging perspective for further understanding cognitive and emotional disturbances related to OSA [24,25]. Several functional magnetic resonance imaging (fMRI) studies suggested that during specific tasks (e.g., visual stimulation and executive functioning tasks), patients with OSA may exhibit different functional brain activity from healthy controls, which may involve multiple regions, such as the prefrontal, parietal, and temporal lobes [26,27,28]. A study of cortical activation patterns during a visuospatial N-back task reveals that both the task-positive network and DMN are affected in individuals with OSA [28]. In addition, the severity of OSA may be associated with neural activation during working memory tasks, suggesting potential compensatory neural responses [26]. A study related to executive function and emotional processing indicates that children’s cognitive control, conflict monitoring, and attention allocation are influenced by OSA [27].

As research progresses, fMRI has demonstrated its importance in examining brain functional connectivity and networks. However, despite its remarkable achievements, fMRI also has some noteworthy drawbacks. Firstly, its high cost makes it a financial consideration in many studies. Additionally, fMRI needs to be conducted in a relatively confined environment, which may impact the comfort and participation of participants. Most importantly, compared to some other brain imaging techniques, fMRI has relatively low temporal resolution, making it challenging to capture detailed dynamic changes in brain function. Functional Near Infrared Spectroscopy (fNIRS), a non-invasive brain imaging technique, offers better temporal resolution and a relatively high tolerance to head and body movements compared to fMRI. Due to its advantages of being lightweight, low-cost, and its ability to be used in relatively natural environments, it has been used to study the underlying pathogenesis of a variety of neurological disorders, such as central nervous system disorders (including AD, depression, etc.). A few studies have successfully used fNIRS to explore brain activity in patients with OSA. To illustrate, Z. Mingming et al. used fNIRS in conjunction with graph theory metrics to assess brain network abnormalities in the prefrontal lobe of patients with OSA [29]. The study observed that patients with OSA exhibited a reduced number of connecting edges between the right central prefrontal cortex and other right hemisphere regions as well as lower global efficiency, local efficiency, and clustering coefficients compared to the HC group [29]. Z. Zhang et al. studied the dynamics of cerebral hemodynamics during nocturnal CPAP therapy [30]. They found that cyclic oscillations in oxyhemoglobin concentration, deoxyhemoglobin concentration, tissue oxygenation index, and blood volume associated with cyclic apneic events were eliminated in all patients with OSA after CPAP treatment [30]. Additionally, S. Baillieul et al. employed fNIRS to assess the effect of CPAP on gait control in patients with severe OSA syndrome [31]. However, their trial revealed that eight weeks of CPAP treatment did not improve gait control in nonobese patients with OSA [31]. These studies demonstrated the feasibility of fNIRS to study brain activity in patients with OSA. Patients with OSA may experience cognitive decline during the daytime, such as diminished attention, memory, and decision-making skills. The temporoparietal lobe plays a key role in processing cognitive functions, such as memory, attention, and perception [7,32]. Studying daytime fNIRS signals in the temporoparietal lobes of patients with OSA is important for understanding the effects of OSA on brain function. Resting-state studies, which are not contingent on subjects performing specific tasks, offer a more reflective portrayal of the brain’s spontaneity. This helps capture the intrinsic activity of the brain without introducing external interference. Therefore, our experiments will be conducted in the resting state. It is noteworthy that no study has used resting-state fNIRS to study the functional connectivity and topological properties of the temporal parietal lobe in patients with OSA.

This study aims to investigate alterations in brain functional connectivity properties in the temporoparietal lobe of patients with OSA using fNIRS techniques. In our study, we utilized resting-state functional connectivity (RSFC) combined with graph theory methods to explore connectivity changes in the temporoparietal regions and analyze the topological properties of the functional network. Subsequently, we compared intergroup differences in RSFC and multiple network parameters in the temporoparietal regions between patients with OSA and healthy controls (HCs). Additionally, we assessed the relationship between abnormal characteristic changes and the AHI score using linear regression. This study examines the influence of OSA from the perspective of brain function, and investigates alterations in brain functional connectivity properties using fNIRS.

## 2. Materials and Methods

### 2.1. Participants

After recruiting subjects, we ensured in the informed consent form that the subjects did not have any mental or neurological disorders before performing the test. In addition to this, we informed the subjects in advance not to consume beverages such as coffee and tea for 12 h prior to the test. During the 10–15 min resting-state fNIRS experiment, participants were asked to remain as still as possible, with their bodies naturally relaxed and eyes closed, and not to fall asleep or to think any particular things. The light and temperature in the laboratory remained constant and comfortable throughout the experiment.

A total of 980 Chinese subjects participated in this experiment in Shanghai, and we used the coefficient of variation to exclude 186 subjects’ data. The CV is expressed as a percentage, calculated by the formula CV (%) = 100 × standard deviation of the data/mean of the data. The signal quality of all individual channels was assessed through the CV, with channels exceeding a certain threshold (CV > 15%) indicating the presence of non-physiological noise and, therefore, being excluded from further processing [33,34,35]. We monitored the subjects’ sleep prior to the experiment and obtained the subjects’ AHI. Subjects with an AHI > 5 were defined as patients with OSA, and those with an AHI < 5 were defined as healthy individuals. Then, from the 792 subjects, two groups were selected that were balanced in terms of gender and age, that is, the group with OSA and the healthy control group. A total of 298 subjects (185 males and 103 females) participated in the data analysis of this study.

The study protocol was approved by the ethics committee of the School of Life Sciences of Fudan University, and informed consent was obtained from all subjects.

### 2.2. fNIRS Recording

A multichannel fNIRS instrument (NirScout, NIRx Medizintechnik GmbH, Berlin, Germany) with a sampling rate of 8.928571 Hz measured relative changes in hemoglobin concentration in the temporoparietal area cortex using infrared light at two wavelengths, 690 and 830 nm. The seven sources and seven detectors formed a total of 16 channels [36]. Each channel consisted of a source–detector pair at a distance of 3–4 cm and was placed with reference to a 10–20 system. Optode placement is shown in Figure 1.

The positions of all NIRS channels were located using a 3D digitizing system that identified the positions of 14 NIRS photodiodes based on the origin and four reference points (Nz, AR, AL, Cz). The coordinates of the fNIRS channels based on source and detector positions were calculated using the MATLAB toolbox AtlasViewer [37]. The estimated average positions of the 16 channels were obtained based on the anatomical information of the Brodmann Areas. The MNI coordinates and label names are showed in Table 1. In this study, the data acquisition used NIRStar software (Version 15.2).

### 2.3. Data Analysis

#### 2.3.1. Preprocessing

The fNIRS data were preprocessed using Matlab package Homer2. Subjects typically required several minutes to transition into a quiet state from the commencement of data acquisition. To ensure a sufficiently reliable signal for analysis, we excluded the initial few minutes of data from the fNIRS dataset and trimmed the final data length to 10 min. The raw light intensity signals were first converted to changes in optical density, after which the processed fNIRS time series were corrected for motion artifacts. Motion artifact correction was followed by filtering out instrumental noise and physiological noise, such as low-frequency drift, heartbeat, and respiration, using bandpass filters with cutoff frequencies of 0.01 Hz and 0.1 Hz, respectively. Subsequently, the preprocessed optical density data at both wavelengths were converted to relative changes in the concentration of oxyhemoglobin (HbO_2_) and deoxyhemoglobin (HbR) by a modified Beer–Lambert law.

#### 2.3.2. Functional Connectivity

RSFC is used to study the functional associations between different regions of the brain when subjects are not performing a specific task and are at rest. Functional connectivity strength is a measure of the strength or interrelationships of connections between different regions in a brain network. For each selected channel, the correlation between its time series and those of all other brain regions was calculated to represent the strength of RSFC. In the present study, only HbO_2_ was used to calculate functional connectivity because a previous study has shown that HbO_2_ is more sensitive to blood flow and oxygen transport than HbR [38]. After calculating the Pearson correlation coefficients between channels, a correlation matrix was generated for each participant’s data.

#### 2.3.3. Graph Theory Analysis

In brain networks, channels are usually defined as nodes and the correlation coefficients between any channel pairs are defined as edges. We used sparsity thresholding to binarize the correlation matrix and compute graph-theoretic metrics. The sparsity S can be expressed as the percentage of connections retained in the network [39]. If S is 0.2, the sorted connections with connection weights in the top 20% will be retained. By adjusting the sparsity, the density of the network can be controlled so that the connections retained in the network are more prominent or important [40,41]. When S is too large, it indicates that the network is relatively dense and retains more connections. This can lead to an overly complex network that contains noise or less important connections. An excessively small S may lead to a loss of information and make the network structure less complete. Using sparsity thresholding, we computed each graph theory parameter along a range of edge densities of 0.2 < S < 0.8 in 0.05 increments. Then, we used MATLAB software (Version R2018b) to compute global and local network properties in binarized networks. Global network properties include global efficiency (Eg), local efficiency (Eloc), clustering coefficient (Cp), shortest path length (Lp), and small-world properties (σ, γ, λ). Regional network properties include nodal betweenness centrality (BC) and degree centrality (DC). A sparsity threshold of around 0.35 preserves important topological information in the network while keeping the network less complex, so we mainly analyzed the network characteristics at a sparsity threshold of 0.35.

#### 2.3.4. Statistical Analysis

Statistical analysis was performed using RStudio (v4.2.3). A simple *t*-test was used for body mass index (BMI) differences. Gender differences were statistically quantified using a chi-square test. Age, AHI, RSFC, and graph-based indexes between the two groups were assessed using the Mann–Whitney U test. Subsequently, we conducted multiple linear regression analyses to assess the potential association between the extracted features and the clinical evaluation AHI. The analysis involved calculating the regression between the z-value of the feature and the AHI, while considering age, BMI, and gender as covariates. Statistical significance for each independent variable was determined with a significance threshold set at *p* < 0.05 after applying the Benjamini–Hochberg (BH) correction.

## 3. Results

### 3.1. Demographic Characteristics

The demographic information of subjects is detailed in Table 2. There was no significant difference between patients with OSA and controls in terms of age (*p* = 0.784) and gender (*p* = 0.451). The BMI (*p* < 0.01) and AHI (*p* < 0.01) were significantly higher in patients with OSA than in the control group. Other indicators of anxiety, depression, and sleep quality were not significantly different between the two groups.

### 3.2. Changes in HbO_2_ Concentration between Patients with OSA and HCs

In the present study, the mean (*p* = 0.023) and standard deviation (*p* = 0.020) of HbO_2_ concentration in Ch14 (the right superior temporal gyrus) were significantly higher in patients with OSA compared to HCs. The mean and standard deviation of hemoglobin concentration in Ch14 were extracted, as shown in Figure 2.

### 3.3. RSFC within the Temporoparietal Network between Patients with OSA and HCs

In both groups, all brain regions showed similar patterns of RSFC. The mean correlation RSFC matrix within the temporoparietal network is shown in Figure 3. The RSFC between Ch1 (the left superior temporal gyrus) and Ch10 (the right precentral gyrus) (*p* = 0.049), and between Ch1 and Ch13 (the right supramarginal gyrus) (*p* = 0.033) were significantly higher in patients with OSA compared to HCs. In addition to this, patients with OSA showed significantly increased RSFC between Ch11 (the right superior temporal gyrus) and Ch10 (*p* = 0.039), and between Ch11 and Ch8 (the left angular gyrus) (*p* = 0.021). Abnormal connections are shown in Figure 4.

### 3.4. Differences in Global Network Metrics of the Temporoparietal Network

Neither patients with OSA nor HCs exhibited an effective small-world topology (σ > 1, γ >> 1, λ ≈ 1) over a wide range of sparsity thresholds defined as 0.2–0.8 (see Figure 5), suggesting that the temporoparietal network does not have typical small-world properties. Cp was higher (*p* = 0.038) and Eloc was higher (*p* = 0.049) in patients with OSA compared to HCs. The differences in Lp (*p* = 0.997) and Eg (*p* = 0.972) were not statistically significant. The global network metrics are illustrated in Figure 6.

### 3.5. Differences in Regional Network Metrics of the Temporoparietal Network

Patients with OSA showed abnormal BC and DC in certain regions. Compared to HCs, BC values were higher in Ch9 (the right precentral gyrus) and Ch13, with higher DC values in Ch13 (*p* < 0.05). Other regions did not show any node metrics that differed between groups. Regional network measures are shown in Table 3.

### 3.6. Correlations between RSFC and AHI

To further investigate the relationship between the temporoparietal network and the diagnostic index of OSA, namely the AHI, linear regression analysis was conducted on selected brain network features and the AHI. Following BH correction, the analysis revealed a significant positive correlation between the AHI and RSFC from Ch10 to Ch11 (coefficient estimate β = 0.017, *p* = 0.040). The negative sign of the regression coefficient for the AHI indicates a direct negative relationship with RSFC between Ch10 and Ch15 (coefficient estimate β = −0.015, *p* = 0.037). This suggests that higher values of the AHI are associated with lower values of RSFC between Ch10 and Ch15.

## 4. Discussion

The aim of this study was to investigate differences in temporoparietal neural activity between patients with OSA and healthy controls. To this end, we compared RSFC as well as graph theory-based network metrics between the two groups. Several studies have shown that there are significant gender and age differences in some brain regions in patients with OSA [42,43,44]. To avoid the influence of these confounders, we selected patients with OSA and healthy individuals with similar age distributions and sex ratios in this study, which can help to confirm that our findings are mainly due to the influence of OSA rather than age or sex differences. To the best of our knowledge, our study is the first resting-state fNIRS study to explore intergroup differences in the temporoparietal lobe between patients with OSA and HCs.

We found that the mean and standard deviation of HbO_2_ concentration in the right superior temporal gyrus were higher in patients with OSA than in healthy individuals. In the resting state, both patients with OSA and healthy people showed regular patterns of changes in blood oxygen concentration, but the right superior temporal gyrus demonstrated an intergroup difference, which indicated that the right and left hemispheres of patients with OSA were differently differentiated. Jing Gao et al. found that the right brainstem, left dorsolateral superior frontal gyrus, and superior temporal gyrus of the group with OSA were atrophied compared to HCs, which suggested that OSA affects the structure of the brain, and the effects are laterally biased [45]. De-Chang Peng et al. found that regional homogeneity was significantly higher in the right posterior cerebellar lobe and right cingulate gyrus in patients with OSA [46]. In our study, the differences in HbO_2_ concentration occurred mainly in the right hemisphere, suggesting that having OSA may lead to significant structural and functional effects on one side of the brain.

The interesting finding in our study was that RSFC between the left superior temporal gyrus and the right precentral gyrus as well as the right supramarginal gyrus was higher in the group with OSA than in the healthy control group. In addition, RSFC between the right superior temporal gyrus and the right precentral gyrus as well as the left angular gyrus was also increased in patients with OSA. The results we found suggest that OSA has a specific effect on RSFC between the superior temporal gyrus and other regions. The superior temporal gyrus plays a variety of important roles in the human brain involving perception, memory, and a number of higher cognitive functions. The superior temporal gyrus is an important part of the default mode network, interconnecting with other core regions of the network. Several studies have shown that increased FC in OSA is commonly seen in the temporoparietal network [7,23,32,47]. The precentral gyrus and supramarginal gyrus are part of the sensorimotor network, which focuses on processing sensory information and motor control. Several studies have shown that OSA selectively impairs RSFC within DMN subregions and also specifically affects RSFC in cognitive and sensorimotor-related brain networks [32,48,49]. The increase in RSFC between the superior temporal gyrus and other regions can be interpreted as a hyper-reflexive or compensatory mechanism under specific conditions and may indicate an adaptive compensatory response of RSFC to cognitive impairment [8].

Several studies have shown that patients with OSA exhibit abnormal small-world organization in either functional or structural brain networks such as the DMN [50,51,52]. However, in the present study, neither patients with OSA nor HCs showed effective small-world organization properties. The small-world networks have a topology with high clustering coefficients and small shortest path lengths, which is not consistent with the results we obtained. The reason is that even though some regions of the temporal and parietal lobes in our study are involved in the DMN, the properties are not exactly equivalent to the DMN. Distinct brain networks may exhibit different properties at different scales. Cp and Eloc are measures of local connectivity and transmission efficiency of the network, respectively. Contrary to the findings of some studies, our results found increased Cp and Eloc in patients with OSA [50,51]. There are also findings in agreement with our results [52]; however, it is worth noting that we studied different regions of the brain network. OSA may lead to changes in brain structure, including changes in gray and white matter. These changes may affect the connectivity patterns of the brain networks and thus the clustering coefficients and local efficiencies. Two opposing patterns may be observed in neuroimaging studies of patients with OSA, one of which may indicate cellular damage while the other may reflect compensatory structural changes [10]. Therefore, the results may vary depending on factors such as the study region or the methodology used.

In the current study, the topological organization of the region between the two groups is compared using BC and DC. DC and BC are two network centrality metrics commonly used in social network analysis, which are used to measure the importance of nodes and their mediations, respectively. Specifically, we observed higher BC and DC in the right supramarginal gyrus and higher DC in the right precentral gyrus in the group with OSA compared with HCs. The consistent trend of increased regional findings may be related to the neuropathologic features of OSA. Our findings generally show more noticeable changes in the right hemisphere of patients with OSA, which is mutually supportive with the results of a previous fMRI study [53]. Similar to the high RSFC in some regions mentioned above, the high BC in the right supramarginal gyrus and the right precentral gyrus may reflect pathological changes or compensatory mechanisms in disease states. Furthermore, since these two regions are components of the default mode network (DMN), our findings support and confirm the perspective of DMN abnormalities in patients with OSA as proposed in previous studies [48,52,54].

The results of the multiple linear regression analysis highlight the impact of the AHI on RSFC between the right precentral gyrus and right superior temporal gyrus, as well as between the right precentral gyrus and right supramarginal gyrus. The AHI, an index used to assess sleep apnea and hypopnea, may be associated with abnormalities inducing physiological and metabolic changes within the brain, thereby affecting functional connectivity among different brain regions [3,22]. Respiratory control and neuromodulation may influence the connections between brain regions, and the onset of sleep disorders could potentially trigger adjustments in these connections. Patients with OSA often experience respiratory control dysfunction, which may be related to morphological differences in certain gray matter regions of the brain [46]. Linear regression results suggest that RSFC between the right precentral gyrus and other brain regions deserves particular attention. This reveals that the function of the right precentral gyrus may be influenced in patients with OSA. The influence received may manifest as changes related to motor control, respiratory regulation, or other neural functions, as the right precentral gyrus is a crucial component of the sensorimotor network [32]. These findings offer a new imaging perspective for a more in-depth understanding of brain functional connectivity disruptions associated with OSA. Significantly, the relative results observed on the right side of the brain underscore a lateralizing effect of OSA on cerebral function.

This study has several limitations. First, the proportion of patients with severe OSA in our study was relatively small, and more patients with severe OSA should be included in future studies. Second, the results of RSFC and network parameters were uncorrected. So, our results should be interpreted with caution and need to be validated by future studies. Compared with previous studies of patients with OSA, we expanded our sample size; however, studies with larger samples are still needed to confirm our results. The chosen focus area for the study is the temporal parietal region, and the research is deemed necessary and lacking in this specific domain, making it worthwhile to investigate OSA pathology in this region. Nevertheless, a comprehensive analysis covering the entire brain may have the potential to reveal deeper insights into the pathology of OSA, encouraging a more thorough exploration of this sleep disorder. Subsequent studies will consider the entire cerebral cortex. In this study, we did not analyze the severity of the cognitive consequences of OSA on individuals and the extent to which OSA affects cognitive function in relation to brain activity patterns, but we plan to conduct such an analysis in the future. The last limitation of this study is the lack of follow-up data. A longitudinal study including pre-treatment data and post-treatment data at different periods is necessary for patients with OSA.

## 5. Conclusions

In summary, the current study examined the resting-state brain function network in patients with OSA. We found that patients with OSA had higher HbO_2_ concentrations in the superior temporal gyrus compared to healthy controls. We also found slightly higher RSFC between the superior temporal gyrus and some other regions, such as the right premolar gyrus, compared to healthy controls. These findings suggest that OSA may lead to abnormal resting-state function in some regions of the brain. Clustering coefficients and local efficiencies of the temporoparietal network in patients with OSA showed an increasing trend. In addition, the betweenness centrality and degree centrality of the right supramarginal gyrus and the meso-centrality of the right precentral gyrus similarly showed an increasing trend. The results of multiple linear regressions demonstrated that the influence of the AHI on RSFC among specific regions in the right temporal and parietal lobes was notably significant. The present study provides some insights into the potential neuropathological mechanisms underlying abnormal brain function in patients with OSA.

## Figures and Tables

**Figure 1 bioengineering-11-00389-f001:**
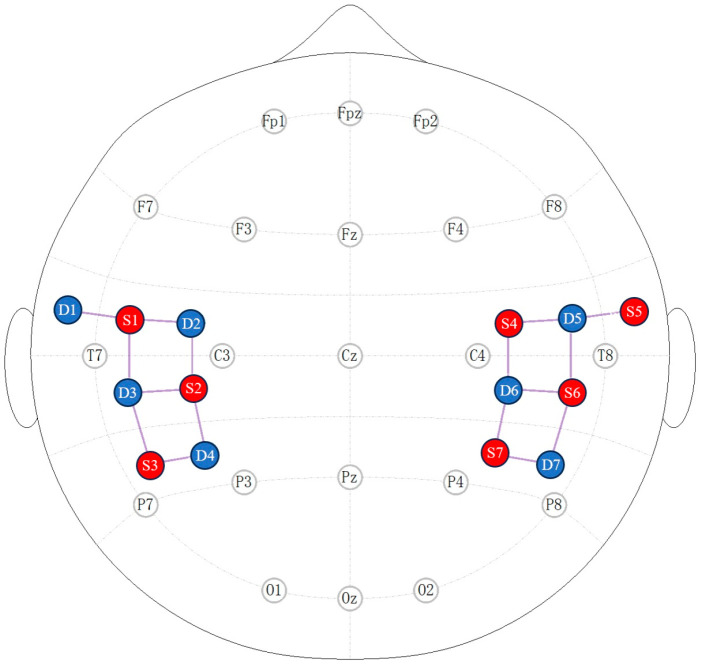
Two-dimensional view of optode placement. Red denotes the source and blue signifies the detector. Seven sources and seven detectors make up a total of 16 channels.

**Figure 2 bioengineering-11-00389-f002:**
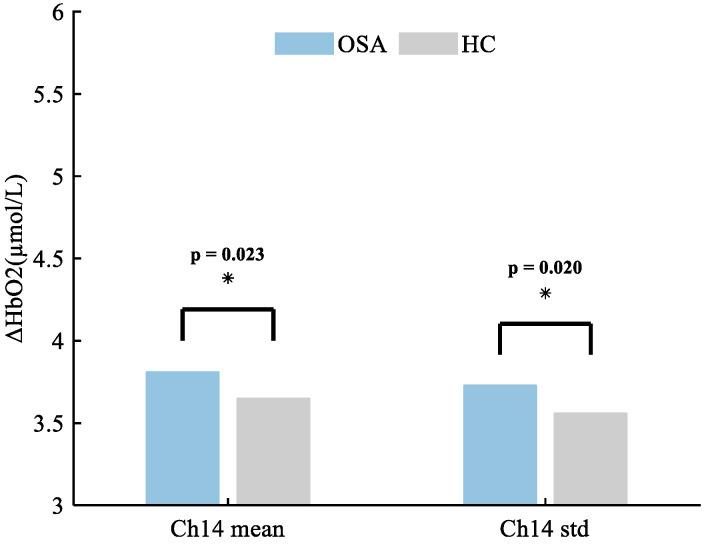
The altered brain regions in patients with OSA exhibit distinctive characteristics in mean and variance of HbO_2_ concentration, with the most prominent changes observed in channel 14, corresponding to the right superior temporal gyrus. Specifically, the mean HbO_2_ concentration demonstrates a significant difference, with values of 3.81 ± 1.54 compared to 3.65 ± 1.51 in individuals without OSA. Additionally, there is a notable distinction in the standard deviation (std) of HbO_2_ concentration, showing values of 3.73 ± 1.03 for patients with OSA versus 3.56 ± 1.53 for HCs. * *p* < 0.05.

**Figure 3 bioengineering-11-00389-f003:**
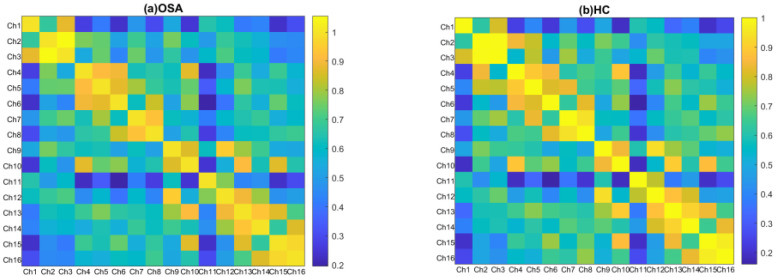
Group-averaged correlation matrix plots of temporoparietal regions for individuals with OSA (**a**) and HCs (**b**). Each pixel in the correlation matrix plot represents the z-value of the Pearson correlation coefficient after Fisher z-transformation for the corresponding channel pair.

**Figure 4 bioengineering-11-00389-f004:**
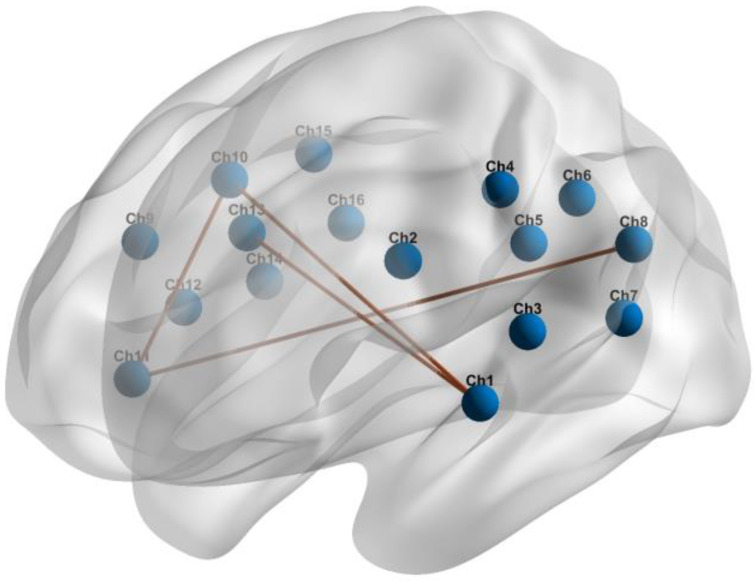
Abnormal RSFC within the temporoparietal brain network in individuals with OSA compared to HCs.

**Figure 5 bioengineering-11-00389-f005:**
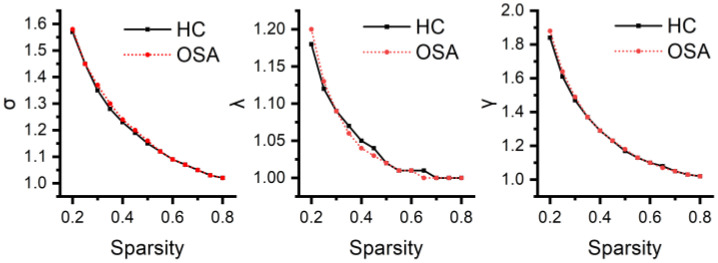
The small-world characteristics of the temporoparietal network for both patients with OSA and HCs across the defined wide threshold range. In the figure, it is evident that the average shortest path length ratio (λ) is approximately equal to one, and the small-world index (σ) is greater than one. However, the clustering coefficient ratio (γ) is not significantly greater than one, indicating that neither patients with OSA nor HCs exhibit a typical small-world topology.

**Figure 6 bioengineering-11-00389-f006:**
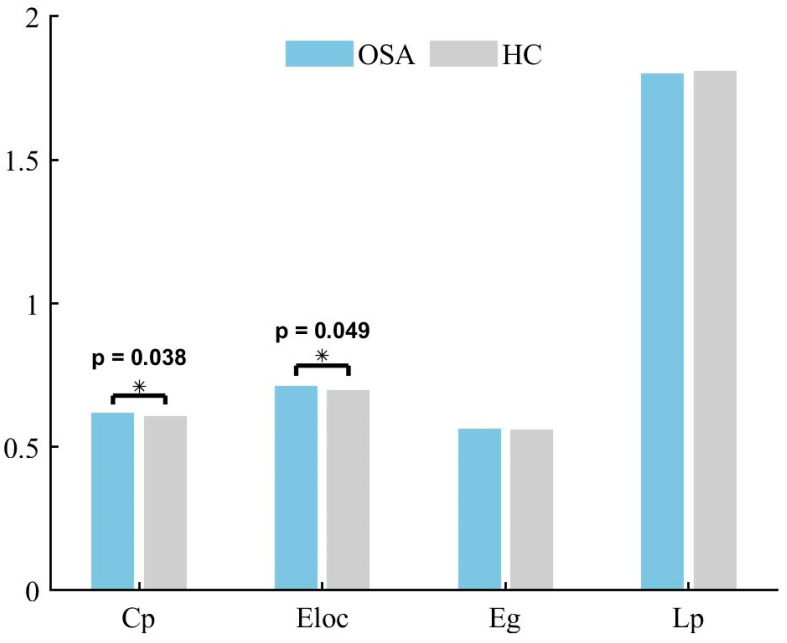
The between-group differences in global network metrics for the temporoparietal network in individuals with OSA and HCs. There is a noticeable trend suggesting a distinction among patients with OSA, with an elevated Cp (*p* < 0.05) and an increased Eloc (*p* < 0.05). * *p* < 0.05.

**Table 1 bioengineering-11-00389-t001:** Channel-projection-to-cortex labels.

Channel	Source	Detector	Region Label	Montreal Neurological Institute (MNI)		
				x	y	z
Ch1	S1	D1	Left superior temporal gyrus	−58	−1	−6
Ch2	S1	D2	Left precentral gyrus	−40	6	24
Ch3	S1	D3	Left superior temporal gyrus	−59	−13	9
Ch4	S2	D2	Left inferior parietal lobule	−48	−14	39
Ch5	S2	D3	Left supramarginal gyrus	−45	−24	27
Ch6	S2	D4	Left supramarginal gyrus	−41	−40	36
Ch7	S3	D3	Left middle temporal gyrus	−59	−39	11
Ch8	S3	D4	Left angular gyrus	−48	−50	26
Ch9	S4	D5	Right precentral gyrus	53	6	24
Ch10	S4	D6	Right precentral gyrus	45	−12	37
Ch11	S5	D5	Right superior temporal gyrus	61	2	−6
Ch12	S6	D5	Right superior temporal gyrus	60	−11	9
Ch13	S6	D6	Right supramarginal gyrus	52	−22	25
Ch14	S6	D7	Right superior temporal gyrus	72	−42	13
Ch15	S7	D6	Right supramarginal gyrus	56	−43	41
Ch16	S7	D7	Right angular gyrus	55	−51	26

**Table 2 bioengineering-11-00389-t002:** Demographic information and AHI of OSA patients and HCs.

Characteristics	Patients with OSA	HCs	*p*-Value
Age, years	42.8 ± 7.91	42.1 ± 11.00	0.784
BMI, kg/m^2^	25.2 ± 3.26	23.2 ± 2.68	<0.01
Gender, M/F	95/53	90/60	0.451
AHI, per hour	2.02 ± 1.75	13.1 ± 9.98	<0.01
Smoking, Y/N	136/12	140/10	0.623
Drinking, Y/N	120/28	117/33	0.530
BAI	25.5	26.2	0.285
BDI	4.95	5.23	0.399
GAD-7	5.28	4.93	0.767
FSS	37.5	37.8	0.816
ESS	8.16	9.10	0.065
PSQI	3.41	3.63	0.530

BAI: Beck Anxiety Inventory Total Score; BDI: Beck Depression Inventory Total Score; GAD-7: Generalized Anxiety Disorder-7 Total Score; FSS: Fatigue Severity Scale Total Score; ESS: Epworth Sleepiness Scale Total Score; PSQI: Pittsburgh Sleep Quality Index Total Score.

**Table 3 bioengineering-11-00389-t003:** The between-group variations in BC and DC of the temporoparietal network among individuals with OSA and HCs.

Channel	Nodal Betweenness Centrality		Nodal Degree Centrality	
	Z Statistic	*p*-Value	Z Statistic	*p*-Value
Ch1	50.067	0.374	47.780	0.160
Ch2	54.853	0.537	57.112	0.206
Ch3	55.861	0.366	55.002	0.508
Ch4	55.344	0.447	53.427	0.831
Ch5	48.620	0.248	51.825	0.808
Ch6	52.691	0.997	54.886	0.529
Ch7	52.286	0.912	54.113	0.683
Ch8	55.214	0.465	56.137	0.324
Ch9	44.251	0.016 *	48.283	0.210
Ch10	51.412	0.719	51.818	0.807
Ch11	54.338	0.567	50.584	0.547
Ch12	51.023	0.638	53.365	0.845
Ch13	45.029	0.030 *	45.487	0.040 *
Ch14	53.695	0.772	52.622	0.988
Ch15	58.372	0.104	52.924	0.944
Ch16	54.497	0.600	54.414	0.621

* *p* < 0.05; the Z statistic is a standardized measure of the U statistic in the Mann–Whitney U test.

## Data Availability

The datasets presented in this article are not readily available because the data are part of an ongoing study.

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
