# Peer review of "Altered Functional Connectivity of Temporoparietal Lobe in Obstructive Sleep Apnea: A Resting-State fNIRS Study"

_bioengineering, 2024, doi:10.3390/bioengineering11040389_

Round 1
Reviewer 1 Report
Comments and Suggestions for Authors
The article presents interesting results on changes in brain connectivity in the patients with Obstructive Sleep Apnea (OSA) . Overall, the manuscript evokes positive impressions. The authors collected a good experimental sample and performed qualitative data analysis. The findings contribute to understanding the causes of behavioral impairment in OSA.
I have some criticisms and recommendations for revising the manuscript before it can be published.
1) My main doubt is that the authors do not analyze the individual severity of the cognitive consequences of the OSA, and the relationship of the degree of the OSA’s influence on cognitive functions with the brain activity patterns. All author-identified differences in the brain activity patterns between the OSA patients and control subjects are weak. A possible reason for the effects weakness is that the OSA causes different consequences for different patients. This can depend on both the individual severity of the disease and many other concomitant factors. In fact, the OCA is not a specific cognitive impairment. Cognitive disorders can be accompanied by the OSA to varying individual degrees. Therefore, different patients may experience different disorders at the level of brain patterns, which weakens the overall effect. In this regard, I recommend either adding additional estimates of the individual severity of cognitive impairment in patients to the analysis, or specifically indicating in the limitation section that such an analysis has not yet been carried out, but is planned in the future.
2) Line 126: Please indicate whether participants were surveyed to assess their anxiety level, general well-being, emotional stress level? Besides, were there comparisons between control groups and patients on their cognitive abilities, such as memory, attention, or motor control?
3) Line 133-135: Please clarify whether the participants' eyes were closed or open during the experiment? What instructions did the participants receive on this matter? Was any control performed on whether participants kept their eyes open or closed?
4) Line 200: Statistical significance of inter-group differences in connectivity was assessed by using the Mann-Whitney U test independently for each pair of cortical regions. This statistical approach poses a threat of multiple comparisons error. In addition to such approach, I recommend using the repeated measures ANOVA with group and topography factors (pairs of cortical structures) adjusted for multiple comparison and subsequent analysis of factor interactions.
When presenting the results of statistical comparisons, I recommend specifying not only the p-level, but also the effect size.
5) Line 222. For Figure 2, I recommend reducing the height of the chart column. The column can start not with a zero value, but with a mark of two, which will make the diagram clearer for understanding. I also recommend specifying the standard errors, for each column of the chart.
Author Response
Response to Reviewer 2 Comments:
Point 2.1: My main doubt is that the authors do not analyze the individual severity of the cognitive consequences of the OSA, and the relationship of the degree of the OSA’s influence on cognitive functions with the brain activity patterns. All author-identified differences in the brain activity patterns between the OSA patients and control subjects are weak. A possible reason for the effects weakness is that the OSA causes different consequences for different patients. This can depend on both the individual severity of the disease and many other concomitant factors. In fact, the OCA is not a specific cognitive impairment. Cognitive disorders can be accompanied by the OSA to varying individual degrees. Therefore, different patients may experience different disorders at the level of brain patterns, which weakens the overall effect. In this regard, I recommend either adding additional estimates of the individual severity of cognitive impairment in patients to the analysis, or specifically indicating in the limitation section that such an analysis has not yet been carried out, but is planned in the future.
Response 2.1: Thanks for the valuable suggestion. Indeed, the consequences of OSA can vary among patients, influenced by the severity of the condition and various other factors. We fully agree that these differences may result in different impairments in brain activity patterns, potentially impacting the robustness of our analyzed results. We have mentioned in the limitations section of the paper that analyses have not yet been conducted to explore individual differences in the cognitive consequences of the severity of OSA and the relationship between these individual differences and brain activity patterns. We recognize that this is a limitation of this study and have explicitly stated this in the paper.
Modifications:
- Revised main text, Page 11, Section Discussion
In this study, we did not analyze the severity of the cognitive consequences of OSA on individuals and the extent to which OSA affects cognitive function in relation to brain activity patterns, but we plan to conduct such an analysis in the future.
Point 2.2: Line 126: Please indicate whether participants were surveyed to assess their anxiety level, general well-being, emotional stress level? Besides, were there comparisons between control groups and patients on their cognitive abilities, such as memory, attention, or motor control?
Response 2.2: Thanks for the valuable suggestion. We have added information on depression, anxiety, fatigue and sleepiness in the demographics section. The BAI is a tool for quantifying the severity of an individual's anxiety symptoms experienced over the past week. The BDI is a widely used tool for assessing the severity of depression symptoms. The GAD-7 is a brief self-report questionnaire used to assess the severity of generalized anxiety disorder (GAD) symptoms over the past two weeks. The FSS is a tool for assessing the severity of fatigue's impact on an individual's daily living. The ESS is a self-administered questionnaire that assesses an individual's general level of daytime sleepiness.The PSQI is an instrument that assesses sleep quality.These scores were not significantly different between the OSA patient group and the control group. And we must candidly report that the current study design did not include a direct assessment of these cognitive abilities in participants. This is mainly because our initial intention was to focus on exploring the association between OSA and resting-state activity patterns of the brain, particularly in daytime fNIRS characteristics. Therefore, the focus of our study was mainly on the collection and analysis of physiological parameters. We appreciate your suggestions, which will help us to conduct a more comprehensive study in our future work. We have clearly mentioned these plans and limitations of the current study in the discussion section of the paper and in the future research directions. We also plan to compare OSA patients and healthy controls in terms of memory, attention, and motor control to gain insight into the potential impact of OSA on cognitive function.
Modifications:
- Revised main text, Page 3, Section 2.1 Participants
Table 2. Demographic information and AHI of OSA patients and HC.
|
Characteristics |
OSA patients |
HC |
p-value |
|
Age,years |
42.8±7.91 |
42.1±11.00 |
0.784 |
|
BMI,kg/m2 |
25.2±3.26 |
23.2±2.68 |
<0.01 |
|
Gender,M/F |
95/53 |
90/60 |
0.451 |
|
AHI,per hour |
2.02±1.75 |
13.1±9.98 |
<0.01 |
|
Smoking,Y/N Drinking,Y/N BAI BDI GAD-7 FSS ESS |
136/12 120/28 25.5 4.95 5.28 37.5 8.16 |
140/10 117/33 26.2 5.23 4.93 37.8 9.10 |
0.623 0.530 0.285 0.399 0.767 0.816 0.065 |
|
PSQI |
3.41 |
3.63 |
0.530 |
BAI: Beck Anxiety Inventory Total Score
BDI: Beck Depression Inventory Total Score
GAD-7: Generalized Anxiety Disorder-7 Total Score
FSS: Fatigue Severity Scale Total Score
ESS: Epworth Sleepiness Scale Total Score
PSQI: Pittsburgh Sleep Quality Index Total Score
Point 2.3: Line 133-135: Please clarify whether the participants' eyes were closed or open during the experiment? What instructions did the participants receive on this matter? Was any control performed on whether participants kept their eyes open or closed?
Response 2.3: Thanks for the valuable suggestion. We have added detailed information in the Methods section of the paper about the state of the participants' eyes closed or open during the experiment, and the specific instructions they received in this regard. We explicitly state that all participants were asked to keep their eyes closed during fNIRS recordings. This was to minimize potential interference from external stimuli and eye movements and to ensure the reliability of the data[1], [2], [3].
Modifications:
- Revised main text, Page 3, Section 2.1 Participants
During the 10-15 min resting-state fNIRS experiment, participants were asked to re-main as still as possible, with their bodies naturally relaxed, eyes closed, and not to fall asleep or to think any particular things. The light and temperature in the laboratory remained constant and comfortable throughout the experiment.
References:
[1] A. Abdalmalak et al., ‘Effects of Systemic Physiology on Mapping Resting-State Networks Using Functional Near-Infrared Spectroscopy’, Front. Neurosci., vol. 16, p. 803297, Mar. 2022, doi: 10.3389/fnins.2022.803297.
[2] R. J. Deligani et al., ‘Electrical and Hemodynamic Neural Functions in People With ALS: An EEG-fNIRS Resting-State Study’, IEEE Trans. Neural Syst. Rehabil. Eng., vol. 28, no. 12, pp. 3129–3139, Dec. 2020, doi: 10.1109/TNSRE.2020.3031495.
[3] W. Sun, X. Wu, T. Zhang, F. Lin, H. Sun, and J. Li, ‘Narrowband Resting-State fNIRS Functional Connectivity in Autism Spectrum Disorder’, Front. Hum. Neurosci., vol. 15, p. 643410, Jun. 2021, doi: 10.3389/fnhum.2021.643410.
Point 2.4: 4. Line 200: Statistical significance of inter-group differences in connectivity was assessed by using the Mann-Whitney U test independently for each pair of cortical regions. This statistical approach poses a threat of multiple comparisons error. In addition to such approach, I recommend using the repeated measures ANOVA with group and topography factors (pairs of cortical structures) adjusted for multiple comparison and subsequent analysis of factor interactions.
When presenting the results of statistical comparisons, I recommend specifying not only the p-level, but also the effect size.
Response 2.4: Thanks for the valuable suggestion. We appreciate your careful review of our analytical methods and your valuable suggestions. In particular, your pointing out of the problem of multiple comparisons is something we recognize as an issue that must be carefully considered when conducting statistical analyses to avoid incorrectly interpreting chance findings as significant results.
In the current study, we did not use the Bonferroni correction or other traditional corrections for multiple comparisons. This decision was based on the following considerations; our analysis focused on specific hypothesis testing rather than exploratory data analysis. As a result, our primary analyses were prespecified and limited to a few key comparisons, reducing the risk of Type I error caused by conducting multiple comparisons at the same time.Methods such as Bonferroni correction, while reducing the risk of false positives, can also significantly reduce the statistical efficacy of a study, especially if the sample size is limited. Given our sample size and the preliminary nature of the study, we are concerned that overly stringent corrections may result in true effects being incorrectly overlooked. Although multiple comparison corrections were not used, we mitigated the effects of the multiple comparison problem in other ways, such as focusing only on comparisons that were biologically significant and supported by prior research, and employing caution when interpreting our findings.
Nonetheless, we recognize the importance of a more formal correction approach when dealing with multiple comparisons and will consider this more carefully in future studies. Our limitations section explains that we did not use a multiple comparison correction and emphasizes that our results should be interpreted with caution and need to be validated by future studies.
Modifications:
- Revised main text, Page 11, Section 4 Discussion
This study has several limitations. First, the proportion of patients with severe OSA in our study was relatively small, and more patients with severe OSA should be included in future studies. Second, the results of RSFC and network parameters were uncorrected. So our results should be interpreted with caution and need to be validated by future studies. Compared with previous studies of OSA patients, we expanded our sample size, however, studies with larger samples are still needed to confirm our re-sults. The chosen focus area for the study is the temporal-parietal region, and the re-search is deemed necessary and lacking in this specific domain, making it worthwhile to investigate OSA pathology in this region. Nevertheless, a comprehensive analysis covering the entire brain may have the potential to reveal deeper insights into the pa-thology of OSA, encouraging a more thorough exploration of this sleep disorder. Sub-sequent studies will consider the entire cerebral cortex. In this study, we did not ana-lyze the severity of the cognitive consequences of OSA on individuals and the extent to which OSA affects cognitive function in relation to brain activity patterns, but we plan to conduct such an analysis in the future. The last limitation of this study is the lack of follow-up data. A longitudinal study including pre-treatment data and post-treatment data at different periods is necessary for OSA patients.
Point 2.5: Line 222. For Figure 2, I recommend reducing the height of the chart column. The column can start not with a zero value, but with a mark of two, which will make the diagram clearer for understanding. I also recommend specifying the standard errors, for each column of the chart.
Response 2.5: Thanks for the valuable suggestion. As per your suggestion, we have adjusted the starting values of the chart columns and the columns are now set to 3 instead of zero.
Modifications:
- Revised main text, Page 2, Section 3.2 Changes in HbO2 concentration between OSA and HC
Reviewer 2 Report
Comments and Suggestions for Authors
My suggestions:
1. No description of the results of studies using NIRS
2. Too little information about the relationship of temporal areas to OSA and the rationale for selecting these areas for the study.
3. The purpose of the study is not indicated.
4. There is no information on where the information qualifying patients for the study and control groups came from.
5. There is no information as to whether people were excluded from the study who had temporal lobe abnormalities that could interfere with the objectivity of the results (by increasing the severity or lack thereof of functional connectivity).
6. It would have been worthwhile to describe the group in more detail (age, place of residence, other diseases: overweight, smoking, vascular diseases, etc.); e.g., brain connections change with age, additional diseases compound the changes; so is it reasonable to qualify all people with such disparate characteristics?
7. There is a lack of information in the data analysis: were all channels taken into account (what was the limiting level of noise?); was all the data from the subjects taken for analysis (was anyone rejected due to a small number of "good" channels?).
Author Response
Point 1.1: No description of the results of studies using NIRS
Response 1.1: Thanks for the valuable suggestion. We summarized the key findings of studies using NIRS and added them to the Introduction section.
Modifications:
- Revised main text, Page 3, Introduction
To illustrate, Z. Mingming et al. used fNIRS in conjunction with graph theory metrics to assess brain network abnormalities in the prefrontal lobe of patients with OSA. They observed that OSA patients exhibited a reduced number of connecting edges between the right central prefrontal cortex and other right hemisphere regions, as well as lower global efficiency, local efficiency, and clustering coefficients compared to the HC group. Z. Zhang et al. studied the dynamics of cerebral hemodynamics during nocturnal CPAP therapy. They found that cyclic oscillations in oxyhemoglobin concentration, deoxyhemoglobin concentration, tissue oxygenation index and blood volume associated with cyclic apneic events were eliminated in all patients with OSA after CPAP treatment. Additionally, S. Baillieul et al. employed fNIRS to assess the effect of CPAP on gait control in patients with severe OSA syndrome. However, their trial revealed that eight weeks of CPAP treatment did not improve gait control in nonobese OSA patients.
Point 1.2: Too little information about the relationship of temporal areas to OSA and the rationale for selecting these areas for the study.
Response 1.2: Thanks for the valuable suggestion. Our study aimed to explore the role of the temporoparietal lobe in patients with OSA, filling a critical research gap in the current fNIRS literature. We have expanded the introduction section to emphasize that that the existing research on OSA's impact on the temporal region is extremely limited, leading to a fragmented understanding of OSA's effects. Focusing on this gap is crucial for gaining a comprehensive understanding of OSA's effects. We highlight the importance of the temporoparietal lobe for understanding cognitive dysfunction in patients with OSA, particularly the role of this region in processing memory, emotion regulation, and auditory information. Despite the relatively limited research currently available on this topic, we provide insights into the relationship between temporal regions and OSA based on our preliminary data or observations. We hope that with these revisions, the paper will more accurately reflect the innovative nature of our study and the new insights it brings to the field.
Modifications:
- Revised main text, Page 3, Introduction
Patients with OSA may experience cognitive decline during the daytime, such as diminished attention, memory, and decision-making skills. The temporoparietal lobe plays a key role in processing cognitive functions such as memory, attention and per-ception[7,32]. Studying daytime fNIRS signal in the temporoparietal lobes of OSA patients is important for understanding the effects of OSA on brain function. Resting-state studies, which are not contingent on subjects performing specific tasks, offer a more reflective portrayal of the brain's spontaneity. This helps capture the intrinsic activity of the brain without introducing external interference. Therefore, our experiments will be conducted in the resting state. It is noteworthy that no study has used rest-ing-state fNIRS to study the functional connectivity and topological properties of the temporal parietal lobe in OSA patients.
References:
[7]M. Caporale et al., ‘Cognitive impairment in obstructive sleep apnea syndrome: a descriptive review’, Sleep Breath, vol. 25, no. 1, pp. 29–40, Mar. 2021, doi: 10.1007/s11325-020-02084-3.
[32] Q. Zhang et al., ‘Altered Resting-State Brain Activity in Obstructive Sleep Apnea’, Sleep, vol. 36, no. 5, pp. 651–659, May 2013, doi: 10.5665/sleep.2620.
Point 1.3: The purpose of the study is not indicated.
Response 1.3: Thanks for the nice suggestion. The main purpose of this study has been added.
Modifications:
- Revised abstract, Page 1
To investigate alterations of brain functional connectivity properties in OSA patients, a comparative analysis of global and local topological properties of brain networks was conducted between OSA patients and healthy controls (HC).
Point 1.4: There is no information on where the information qualifying patients for the study and control groups came from.
Response 1.4: Thanks for the valuable suggestion. We have enhanced the Methods section by including a comprehensive description of the subjects' information. This encompasses details on participant sourcing, criteria for judgment, and any pertinent screening processes. Additionally, demographic data, including AHI, gender, and age, has been incorporated into the Results section.
Modifications:
- Revised main text, Page 4, Section 2.1 Participants
A total of 980 Chinese subjects participated in this experiment in Shanghai, and we used the coefficient of variation (CV) to exclude 186 subjects' data. CV is expressed as a percentage, calculated by the formula CV (%) = 100 × standard deviation of the data / mean of the data Signal quality of all individual channels is assessed through CV, with channels exceeding a certain threshold (CV > 15%) indicating the presence of non-physiological noise and therefore being excluded from further processing[33],[34],[35].We monitored the subjects' sleep prior to the experiment and obtained the subjects' AHI. Subjects with an AHI >5 were defined as OSA patients, and those with an AHI <5 were defined as healthy individuals. Then, from the 792 subjects, two groups were selected that were balanced in terms of gender and age, that is, the OSA patient group and the healthy control group. A total of 298 subjects (185 males and 103 females) participated in the data analysis of this study.
References:
[33] S. B. ErdoÄŸan, M. A. Yücel, and A. Akın, ‘Analysis of task-evoked systemic interference in fNIRS measurements: Insights from fMRI’, NeuroImage, vol. 87, pp. 490–504, Feb. 2014, doi: 10.1016/j.neuroimage.2013.10.024.
[34] L. Hocke, I. Oni, C. Duszynski, A. Corrigan, B. Frederick, and J. Dunn, ‘Automated Processing of fNIRS Data—A Visual Guide to the Pitfalls and Consequences’, Algorithms, vol. 11, no. 5, p. 67, May 2018, doi: 10.3390/a11050067.
[35] O. Seidel, D. Carius, J. Roediger, S. Rumpf, and P. Ragert, ‘Changes in neurovascular coupling during cycling exercise measured by multi-distance fNIRS: a comparison between endurance athletes and physically active controls’, Exp Brain Res, vol. 237, no. 11, pp. 2957–2972, Nov. 2019, doi: 10.1007/s00221-019-05646-4.
Point 1.5: There is no information as to whether people were excluded from the study who had temporal lobe abnormalities that could interfere with the objectivity of the results (by increasing the severity or lack thereof of functional connectivity).
Response 1.5: Thanks for the nice suggestion. We greatly appreciate your careful review and important comments regarding the possibility that temporoparietal lobe abnormalities may affect the objectivity of study results. Indeed, temporoparietal lobe abnormalities may have a significant impact on the results of a study, especially in studies that explore the relationship between temporoparietal lobe function and OSA. However, our recruited subjects primarily targeted the healthy general population and individuals with suspected symptoms of OSA, none of whom had brain disorders or neurological diseases in the temporoparietal region. We have reviewed and updated the Methods section to clearly state the exclusion criteria used in the screening process for all participants.
Modifications:
- Revised main text, Page 3, Section 2.1 Participants
After recruiting the subjects, we ensured in the informed consent form that the subjects did not have any mental or neurological disorders before performing the test. In addition to this, we informed the subjects in advance not to consume beverages such as coffee and tea for 12 hours prior to the test.
Point 1.6: It would have been worthwhile to describe the group in more detail (age, place of residence, other diseases: overweight, smoking, vascular diseases, etc.); e.g., brain connections change with age, additional diseases compound the changes; so is it reasonable to qualify all people with such disparate characteristics?
Response 1.6: Thanks for the valuable suggestion. Our study enrolled participants from the healthy population and those suspected of having Obstructive Sleep Apnea (OSA), specifically excluding individuals with other conditions, such as cardiovascular diseases. We added a detailed description paragraph in the demographic information section of the results, specifying the mean age of the participants, the gender ratio, and other relevant health conditions (e.g., BMI, smoking status, and sleep quality scores). Such a detailed description helps the reader to better understand the representativeness of the study sample and the generalizability of the findings.
Modifications:
- Revised main text, Page 6, 3.1. Demographic characteristics
Table 2. Demographic information and AHI of OSA patients and HC.
|
Characteristics |
OSA patients |
HC |
p-value |
|
Age,years |
42.8±7.91 |
42.1±11.00 |
0.784 |
|
BMI,kg/m2 |
25.2±3.26 |
23.2±2.68 |
<0.01 |
|
Gender,M/F |
95/53 |
90/60 |
0.451 |
|
AHI,per hour |
2.02±1.75 |
13.1±9.98 |
<0.01 |
|
Smoking,Y/N Drinking,Y/N BAI BDI GAD-7 FSS ESS |
136/12 120/28 25.5 4.95 5.28 37.5 8.16 |
140/10 117/33 26.2 5.23 4.93 37.8 9.10 |
0.623 0.530 0.285 0.399 0.767 0.816 0.065 |
|
PSQI |
3.41 |
3.63 |
0.530 |
BAI: Beck Anxiety Inventory Total Score
BDI: Beck Depression Inventory Total Score
GAD-7: Generalized Anxiety Disorder-7 Total Score
FSS: Fatigue Severity Scale Total Score
ESS: Epworth Sleepiness Scale Total Score
PSQI: Pittsburgh Sleep Quality Index Total Score
Point 1.7: There is a lack of information in the data analysis: were all channels taken into account (what was the limiting level of noise?); was all the data from the subjects taken for analysis (was anyone rejected due to a small number of "good" channels?).
Response 1.7: Thanks for the valuable suggestion. For the inclusion of subject data, we now detail the criteria for determining which subjects' data are used in the final analysis. Ultimately, all subjects’ data had good channel data quality. Our analytical approach employed the coefficient of variation (CV) as a critical metric for assessing the quality of signals from all individual channels. We set a specific threshold, wherein channels with a CV exceeding 15% were identified as containing non-physiological noise and were consequently excluded from the subsequent analysis. This CV threshold was diligently applied to ensure that only data of the highest quality were considered, minimizing the impact of noise on our findings. If any subject had channels where the CV exceeded the 15% threshold, indicating a substantial level of non-physiological noise, the entire dataset from that subject was excluded from further analysis. This exclusion criterion was applied to maintain the integrity and reliability of our study's outcomes, ensuring that our analysis was conducted on data sets that meet our stringent quality standards.
Modifications:
- Revised main text, 4, Section 2.1 Participants
A total of 980 Chinese subjects participated in this experiment in Shanghai, and we used the coefficient of variation to exclude 186 subjects' data. CV is expressed as a percentage, calculated by the formula CV (%) = 100 × standard deviation of the data / mean of the data Signal quality of all individual channels is assessed through CV, with channels exceeding a certain threshold (CV > 15%) indicating the presence of non-physiological noise and therefore being excluded from further processing [33],[34],[35]. We monitored the subjects' sleep prior to the experiment and obtained the subjects' AHI. Subjects with an AHI >5 were defined as OSA patients, and those with an AHI <5 were defined as healthy individuals. Then, from the 792 subjects, two groups were selected that were balanced in terms of gender and age, that is, the OSA patient group and the healthy control group. A total of 298 subjects (185 males and 103 females) participated in the data analysis of this study.
References:
[33] S. B. ErdoÄŸan, M. A. Yücel, and A. Akın, ‘Analysis of task-evoked systemic interference in fNIRS measurements: Insights from fMRI’, NeuroImage, vol. 87, pp. 490–504, Feb. 2014, doi: 10.1016/j.neuroimage.2013.10.024.
[34] L. Hocke, I. Oni, C. Duszynski, A. Corrigan, B. Frederick, and J. Dunn, ‘Automated Processing of fNIRS Data—A Visual Guide to the Pitfalls and Consequences’, Algorithms, vol. 11, no. 5, p. 67, May 2018, doi: 10.3390/a11050067.
[35] O. Seidel, D. Carius, J. Roediger, S. Rumpf, and P. Ragert, ‘Changes in neurovascular coupling during cycling exercise measured by multi-distance fNIRS: a comparison between endurance athletes and physically active controls’, Exp Brain Res, vol. 237, no. 11, pp. 2957–2972, Nov. 2019, doi: 10.1007/s00221-019-05646-4.
Round 2
Reviewer 1 Report
Comments and Suggestions for Authors
The manuscript was substantially improved. I recommend that it be accepted for publication in the current form.